# Comparison of a Novel Polymeric Hollow Fiber Heat Exchanger and a Commercially Available Metal Automotive Radiator

**DOI:** 10.3390/polym13071175

**Published:** 2021-04-06

**Authors:** Tereza Kroulíková, Tereza Kůdelová, Erik Bartuli, Jan Vančura, Ilya Astrouski

**Affiliations:** 1Heat Transfer and Fluid Flow Laboratory, Faculty of Mechanical Engineering, Brno University of Technology, Technicka 2, 616 69 Brno, Czech Republic; Tereza.Kudelova@vut.cz (T.K.); Erik.Bartuli1@vut.cz (E.B.); Ilya.Astrouski@vut.cz (I.A.); 2Institute of Automotive Engineering, Faculty of Mechanical Engineering, Brno University of Technology, Technicka 2, 616 69 Brno, Czech Republic; Jan.Vancura@vutbr.cz

**Keywords:** polymeric heat exchanger, hollow fibers, engine cooling, plastic car radiator

## Abstract

A novel heat exchanger for automotive applications developed by the Heat Transfer and Fluid Flow Laboratory at the Brno University of Technology, Czech Republic, is compared with a conventional commercially available metal radiator. The heat transfer surface of this heat exchanger is composed of polymeric hollow fibers made from polyamide 612 by DuPont (Zytel LC6159). The cross-section of the polymeric radiator is identical to the aluminum radiator (louvered fins on flat tubes) in a Skoda Octavia and measures 720 × 480 mm. The goal of the study is to compare the functionality and performance parameters of both radiators based on the results of tests in a calibrated air wind tunnel. During testing, both heat exchangers were tested in conventional conditions used for car radiators with different air flow and coolant (50% ethylene glycol) rates. The polymeric hollow fiber heat exchanger demonstrated about 20% higher thermal performance for the same air flow. The efficiency of the polymeric radiator was in the range 80–93% and the efficiency of the aluminum radiator was in the range 64–84%. The polymeric radiator is 30% lighter than its conventional metal competitor. Both tested radiators had very similar pressure loss on the liquid side, but the polymeric radiator featured higher air pressure loss.

## 1. Introduction

Polymeric heat exchangers (HXs) are used for special purposes, typically in industrial chemical applications and for use with highly corrosive surroundings. Commercially available HXs are typically made of polymeric tubes with a diameter of 5–50 mm and a wall thickness of 1–3 mm. The polymeric HXs discussed in this paper are very different, where their heat transfer surface is composed of polymeric hollow fibers with a maximum outer diameter of 1 mm and wall thickness of about 0.1 mm. In such a HX, the heat transfer surface is made of several thousand of these tiny tubes. This type of HX was first discussed by Zarkadas in 2004 [1]. Significant efforts have been made since 2004 to study the different properties of this type of heat transfer surface. It has been found that thousands of flexible hollow fibers must be properly separated and distributed in space to form an efficient heat transfer surface. If the fiber structure is not well-organized, there are large performance losses due to flow dead zones and bypasses. Several separation techniques for planar HXs have been described in [2]. Heat transfer surfaces can also be formed by assembling a bundle of fibers. In this case, fiber separation is difficult. A special bundle chaotization procedure is described in [3]. Each fiber in this chaotic bundle has a unique shape and the fibers only have pointwise contacts between each other. For liquid–liquid shell and tube HXs, some kind of plastic net (one or more) can be installed to hold the fibers in a given position [4,5]. The overall heat transfer coefficient of the separated module is 30% higher than that of a HX without a net.

The smooth surfaces of the extruded fibers provide very good resistance to fouling. Fouling has been studied both for air–liquid HXs with dust in the air [6] and for liquid–liquid HXs with waste water [7]. It has been shown that the fouling on plastic tubes is about four times slower when compared to fouling on louvered fins of a similar size [8]. In addition, fouling deposits can easily be cleaned from a polymeric fiber surface due to its smoothness and weak adhesion forces.

The fact that HXs made completely of polymeric materials are nonconductive is another positive factor for electrical devices. The use of these HXs for computer cooling was discussed in [9]. Possible applications in electric vehicles as cooling and heating devices are discussed in [10,11], where the contact cooling of clusters of cylindrical cells was studied experimentally. They demonstrated very high performance and it was discovered that a very uniform constant cell temperature could be maintained.

The use of HXs with hollow fibers has also been investigated in the context of desalination, where corrosion resistance is essential. In Song’s study [12], the properties of HXs composed of fibers made from PP (polypropylene), PES (polyester), and PEEK (polyetheretherketon) materials were tested. Cross flow HXs were constructed and an overall heat transfer coefficient of up to 2000 W/m^2^K was found for the laboratory PP prototypes in terms of a liquid–liquid application. The authors of [13] also consider hollow fiber HXs to be very good substitutes for heat recovery systems in building applications. It was concluded in [14] that polymeric hollow fiber heat exchangers (PHFHE) can be used as evaporators with an overall heat transfer coefficient over 2000 W/m^2^K when the fiber wall is less than 0.1 mm.

Studies on the application of hollow fiber surfaces for gas–liquid heat transfer are not widespread. There are studies which have investigated heat transfer on banks of small polymeric tubes, for example [15,16]. Other studies have focused on improving the properties of polymeric material in order to enhance thermal conductivity and make it possible to use finned tubes [17]. The authors of [18] compared the characteristics of circular, oval, and teardrop tube shapes and proposed optimal geometrical parameters. As plastic pipes are produced by extrusion and can be easily shaped alongside the extrusion axis, this approach can easily be applied.

The possible use of heat exchangers with hollow fibers in car radiators was discussed by Krasny [19]. The conclusions drawn in the paper were made based on experiments with smaller HXs with polypropylene fibers with outer diameters of 0.6 and 0.8 mm and a frontal cross-section of 250 × 250 mm. It was shown that the existing heat transfer theory can be used for estimating the performance parameters. It was found that these devices can achieve high overall heat transfer coefficients (up to 335 W/m^2^ K) and efficiencies (up to 0.8).

In this paper, the authors present a laboratory prototype of a polymeric hollow fiber and comparison with a commercially available aluminum radiator. The goal of the study is to determine the strengths and weaknesses of the PHFHE and the possibility of replacing the conventional radiator with the new radiator. We focus on the thermal performance and pressure drop of the novel HX and compare it with the existing HX. These are very important characteristics for an automotive radiator and it is necessary to know them for further development of a polymeric hollow fiber radiator.

## 2. Materials and Methods

### 2.1. Heat Transfer and Pressure Drop of Polymeric Hollow Fibers

The main advantage of a heat transfer surface consisting of hollow fibers in comparison to metal parts in heat exchangers is the high heat transfer coefficient. This is typical for heat transfer from micro-sized surfaces. Fluid flow inside the fibers is laminar and therefore the Nusselt number is constant and does not depend on velocity. Thus, a very high heat transfer coefficient can be obtained even at low velocities.

The fiber wall thickness is typically about 0.1 mm, and the thermal resistance of the fibers is equivalent to the thermal resistance of aluminum and copper tubes used in metal HXs.

#### 2.1.1. Fluid Flow and Heat Transfer Inside the Fiber

Hickman’s approach [1] gives a solution for an incompressible fluid flowing through a circular tube (maintained at a constant temperature) with constant physical properties, a fully developed laminar velocity profile, and a thermally developing temperature profile. The third type of convective boundary condition is utilized; this assumes that the outside wall temperature of the circular duct is axially constant and that the heat flux is linearly proportional to the temperature difference between the outside wall temperature and the inside wall temperature, which can vary axially as well as peripherally. The following equation was proposed to determine the asymptotic Nusselt number:(1)$${\mathrm{Nu}}_{T3}=\frac{\left(48/11\right)+{\mathrm{Nu}}_{w}}{1+\left(59/220\right){\mathrm{Nu}}_{w}}.$$

Equation (1) yields ${\mathrm{Nu}}_{T3}$ values which fall between 3.66 and 4.36. The lower limit of this Nusselt number range is the asymptotic Nusselt number corresponding to the constant wall temperature boundary condition $\left({\mathrm{Nu}}_{w}=\infty \right)$ and the upper limit is the limiting Nusselt number corresponding to the constant heat flux boundary condition $\left({\mathrm{Nu}}_{w}=0\right)$. This simplified solution does not take into account the influence of a developing flow region and underestimates heat transfer for short ducts but is sufficient for long fibers (L/Di > 200). The computed heat transfer coefficient (HTC) inside the hollow fibers with an internal diameter of 0.64 mm is 2760 W/m^2^ K.

The hollow fibers can be considered long smooth microchannels and the conventional Poiseuille number Po = 64 can be used to calculate the pressure drop; however, the temperature and liquid viscosity vary with a changing fiber length and this must be considered. Thus, the pressure drop (along the fiber, due to the viscous friction, excluding pressure drops of the fiber inlet/outlet) can be expressed as:(2)$$\Delta p_{it}=\frac{128\;\;\mu_{av}\;\;l\;\;Q_{f,t}}{\pi \;\;D_{i}^{4}\;\;N},$$
where $Q_{f,t}$ is the volumetric flow rate through the fiber, l is the fiber length, Di is the internal diameter, N is the number of fibers, and $\mu_{av}$ is the average value of viscosity along the fiber [20].

We should consider not only the pressure drop along the fiber due to viscous friction, but also the local pressure drops of the inlet and the outlet of the fiber. These pressure drops can be determined using conventional approaches; however, calculations show that they are relatively low compared to friction pressure drops along the fiber (1–2% of the total).

Figure 1 shows that the friction pressure drops, according to Equation (2), in which hollow fibers are strongly influenced by the temperature due to the viscosity change of the ethylene glycol (EG) and water solution with temperature. This is important and advantageous in the use of this type of heat transfer surface in a car radiator.

#### 2.1.2. Air Flow Across a Bank of Fibers

Considering the flow along the outer surface of the hollow fiber bunch, the Nusselt number for a given geometry can be expressed in terms of the Reynolds number and the Prandtl number. To predict the Nusselt number, authors usually use the following two empirical models. The first model is for a single tube and the second is for a bank of tubes within a crossflow of air. The average Nusselt number for a cross-flow over a single tube can be evaluated using the equation developed by Churchill and Bernstein [21]:(3)Nud=0.3+0.62Re1/2Pr1/31+(0.4∕Pr)2/31/41+Re282.0005/84/5.

For a bunch of tubes, depending on fiber width and depth pitches, the Grimison approach can be recommended [22]:(4)Nu=1.13C Rem Pr13,
where experimentally determined constants C and exponents m  represent a geometric description of the tube-bundle arrangement. For the geometry used in this paper, the constants C = 0.452 and m = 0.568 are suitable.

By using Equations (1) and (4) and the thermal resistance of wall we can describe how the overall heat transfer coefficient increases as the diameter of the hollow fiber decreases. Figure 2 shows the computed overall heat transfer coefficient for a layer of fibers in an air crossflow. This example considers polyamide fibers with a wall thickness of 10% of the outside diameter. It is obvious that from the heat transfer point of view it is better to use small fibers. The effect of heat transfer in the design of HXs must be balanced with the demands on the liquid pressure drop, and the use of fibers with a very small diameter requires an enormous number of fibers in HXs.

The following equation was used for the sake of computation efficiency:(5)$$E=\frac{T_{Air,Out}-T_{Air,In}}{T_{Liq,In}-T_{Air,In}},$$
where the meanings of the symbols are obvious and the temperatures were taken from the measurement in the calorimetric tunnel.

### 2.2. Dimensional and Mass Comparison of Polymeric and Metal Radiators

A new type of polymeric heat exchanger for automotive applications was developed and constructed by the Heat Transfer and Fluid Flow Laboratory at the Brno University of Technology, Czech Republic. The polymeric hollow fiber heat exchanger consists of 12,240 fibers with a length of 480 mm. The polymeric hollow fibers were produced in the laboratory via a hotmelt extrusion of granules of high viscose polyamide 612 produced by DuPont (Zytel LC6159, Wilmington, DE, USA). This material was chosen because of the relative ease of extrusion to form a uniform and high-quality wall in the final product. Also, this material has a relatively high melting point of 216 °C and the temperature of deflection under a load of 0.45 MPa is 150 °C. This means that even at this temperature the wall still retains its strength characteristics. This is very important for this work since the temperature of the working fluid can reach 120 °C. The outer fiber diameter was 0.8 mm and the inner diameter was 0.64 mm. A 5% variation in the diameter is common due to extrusion. As the wall thickness was small (about 0.08 mm) it was not necessary to use more expensive polymers with higher thermal conductivity for the air–water application. The main thermal resistance in this case will occur at the wall–air interface, and thus the thermal conductivity of the wall does not play a significant role in heat transfer [3]. The area of the heat transfer surface was 12.92 m^2^. The fibers were distributed in 34 layers of 360 fibers. The fibers in the layer were woven together using a textile fiber and polypropylene tubing with an outer diameter of 0.6 mm (detailed in Figure 3) which was attached perpendicularly to the active hollow fibers. This polypropylene tubing also provided spacing between the layers of the heat exchanger. The frame of the heat transfer core and manifolds was made of plastic reinforced with glass fiber. The assembled polymeric car radiator is shown in Figure 4.

The polymeric HX was designed and built to replace the metal radiator (see Figure 5) in a Skoda Octavia (Skoda, Mladá Boleslav, Czech Republic) with a 3rd generation 1.4 L turbocharged stratified injected (TSI) gasoline engine with a maximum power of 110 kW. A photo of the heat transfer surface of the tested radiator from Octavia is shown in Figure 6. The heat transfer surface is made of sinusoidal louvered aluminum fins that were brazed to flat aluminum tubes in an inert atmosphere.

Table 1 shows the weight of the heat transfer surface without and with coolant, which was weighed in the laboratory, as well as the geometrical details of the primary elements of the heat transfer surface. The weights are for the heat transfer surface only, without the inlet and outlet chambers. The heat transfer surface of the polymeric HX has a lower weight. Even though the polymeric one holds more coolant, the total weight of the heat transfer surface and the coolant is still lower than for a conventional aluminum radiator.

### 2.3. Experimental Details

Both heat exchangers were tested at the accredited calorimetric circuit (RAIV s.r.o., Liberec, Czech Republic). This circuit using two sets of nine thermocouples, first set is in front of the HX and latter set is behind HX. This allows the measurement of air flow temperature on both side of the HX during experimentation. Also, it is possible to measure pressure drops on the air/coolant side and the coolant temperature. The heat rejection of the heat exchanger on the air and coolant sides was computed using this data.

A mixture of 50% water and 50% ethylene-glycol (Chevron, San Ramon, USA) was used as a radiator coolant for testing. The input liquid temperature was 90 °C and the intake air temperature was 30 °C. The tests of the polymer radiator were carried out with air velocities of 1, 2, 3, and 4 m/s and liquid flow rates of 15, 30, 45, and 60 L/min. The tests with the metal radiator were carried out with air velocities of 2, 4, and 6 m/s and liquid flow rates of 30, 60, and 90 L/min. Installation of the polymeric radiator in the test bench is shown in Figure 7.

The thermal outputs were computed based on results measured on both the air and the liquid sides. The thermal balance errors obtained from the measurements of the air side and liquid side were less than 3%. Figure 8 shows the surface temperature distribution of the polymer radiator during testing. It is obvious that the inlet manifold is located on the top. The inlet and outlet in the used prototype are shown on the left side of the radiator and it was evident that the fibers on the left side of the radiator featured a higher temperature. A few fibers in the right part of the figure were cold (blue area) which indicated that they were blocked during prototype production.

## 3. Results and Discussion

The performance and pressure drop data taken from the experiments with both heat exchangers are presented below. The performance parameters depended on the air velocity through the radiator as shown in Figure 9. Data are shown for a coolant flow rate of 60 L/min. It can be seen that for an air velocity higher than 2 m/s, the PHFHE has 25% better thermal performance than the metal HX. With an increase in air velocity to 4 m/s, the difference in thermal performance between the PHFHE and metal HX increases to almost 30%.

A comparison of the influence of thermal performance rate on the coolant flow rate in both radiators is shown in Figure 10, where the results for air velocities of 2 m/s and 4 m/s are shown. The graph shows that for an air velocity of 2 m/s and for liquid flow rates less than 40 L/min that the PHFHE has better thermal performance than the metal radiator, even for an air velocity which is twice as high, i.e., 4 m/s. The maximal thermal performance for the PHFHE was 70 kW with a liquid flow rate of 60 L/min and air velocity of 4 m/s.

Figure 11 shows that the hydraulic characteristics on the liquid side for both radiators were similar. Hydraulic losses should be discussed carefully. The results shown in Figure 11 represent identical inlet liquid temperatures of 90 °C for both radiators. It can be seen that the PHFHE’s curve is flatter. As mentioned previously, this is because there is always laminar water flow inside the hollow fiber for this condition. For laminar liquid flow, dependence between the pressure loss and the flow rate was linear. A slight turbulent behavior of the liquid flow is achieved only in the inlet and outlet header of the heat exchanger, which explains its larger cross-section. The mostly laminar nature of the liquid flow inside the PHFHE leads to the fact that at a higher flow rate, the pressure drop of metal HX would be higher than that of polymer HX.

On the other hand, the pressure losses on the air side were very different (see Figure 12) as the polymeric radiator was denser. The air-side pressure drop for the PHFHE was as much as six times higher than that of its metal counterpart. Since the PHFHE has a higher heat transfer rate, it is only possible to remove a few fiber rows. This should reduce the air-side pressure drop, but it is hard to predict by how much since the empirical relationships for a bank of tubes in cross-flow are not clear in the case of flexible polymeric hollow fibers [23].

Higher hydraulic resistance on the air side in combination with higher efficiency can be potentially advantageous in automobiles as less air blowing through the radiator reduces the drag coefficient of the automobile.

A comparison of the efficiencies of the two radiators according to Equation (5) is shown in Figure 13. The graph shows that the efficiency of the PHFHE was higher than that of the metal radiator and did not fall below 80% for the entire measurement range.

The presented results from the wind tunnel tests shows good thermal performance for the polymeric radiator when compared with the metal one. Coolant flow inside the fibers was laminar, presenting the advantage of the linear (not parabolic) dependence of pressure loss on the flow rate. As mentioned previously, the result of this is slower growth for pressure loss with an increasing flow rate, which is the case with an aluminum HX, which has larger channels and the flow is therefore turbulent with the same liquid flow rate. Figure 11 shows that the function is not completely linear in whole PHFHE due to presence of turbulent flow in the input/output chambers. Also, the strong dependence of the pressure losses inside the fibers on fluid viscosity is very important (see Figure 1).

The heat transfer surface weight for the PHFHE is 40% with an aluminium HX heat transfer surface. As the PHFHE has a larger inside volume, the heat exchanger filled with coolant makes up 70% of the weight of an aluminium HX. This is still a very good result that demonstrates that polymer heat exchangers can be 30% lighter than their metal counterparts and also feature comparable heat transfer performance. To further improve polymeric heat exchangers, it is necessary to optimize the amount and distribution of fibers in the HX to reducing the pressure loss on the air side.

## 4. Conclusions

The present study has provided a direct comparison of different heat transfer surfaces used in car radiators. The novel polymeric heat exchanger uses hollow polyamide fibers with diameters of 0.8/0.64 mm and the commercially available metal radiator uses an aluminum heat transfer surface formed by louvered fins brazed to flat tubes.

The tested polymeric heat exchanger was 30% lighter than the metal one (data with tubes filled with coolant). The pressure drop testing showed very similar hydraulic characteristics on the liquid side for both radiators. The inside flow of the coolant in the hollow fibers was laminar and this presents two main advantages. The first is that the heat transfer coefficient is high and independent from the flow velocity. The second is that the pressure losses inside the fibers grow linearly with velocity due to the laminar flow, while in the metal radiator the growth is parabolic (faster) due to the turbulent flow.

The efficiency of the PHFHE was in the range of 80–93% and the efficiency of the metal HX for identical parameters was in the range of 64–84%. The maximum heat transfer performance of the PHFHE was 30% higher than the metal one and reached a value of 70 kW. Unfortunately, the polymeric HX is “denser” and featured higher pressure losses on the air side. This can be solved by further optimizing the heat exchanger, for example by reducing the number of fibers in the heat exchanger.

The polymeric radiator appears to be an alternative to a commercial aluminum radiator in terms of performance. The heat transfer surface is significantly lighter, which would result in lower fuel consumption and CO_2_ emissions. Also, the benefits of a polymeric material, such as a lower melting temperature, would save on manufacturing costs in mass production.

## Figures and Tables

**Figure 1 polymers-13-01175-f001:**
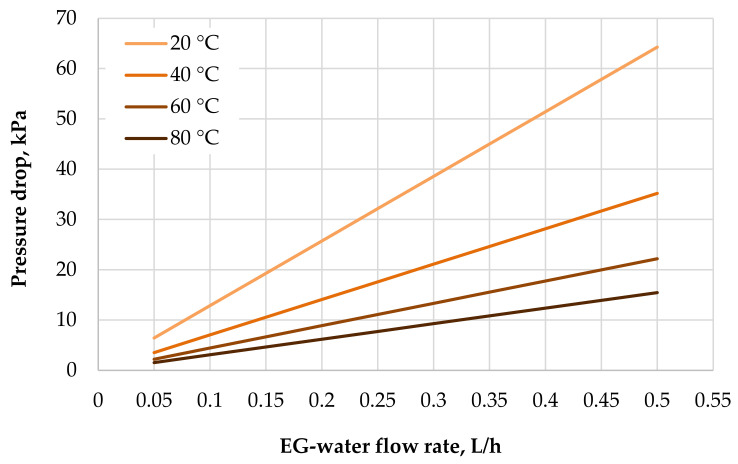
Friction pressure drops vs. flow rate of 50% EG (ethylene glycol) and water flowing in fiber (0.05–0.5 L/h solution flow rate, solution temperature 0–80 °C, fiber inner diameter 0.64 mm, 500-mm-long single fiber). Data computed according to Equation (2).

**Figure 2 polymers-13-01175-f002:**
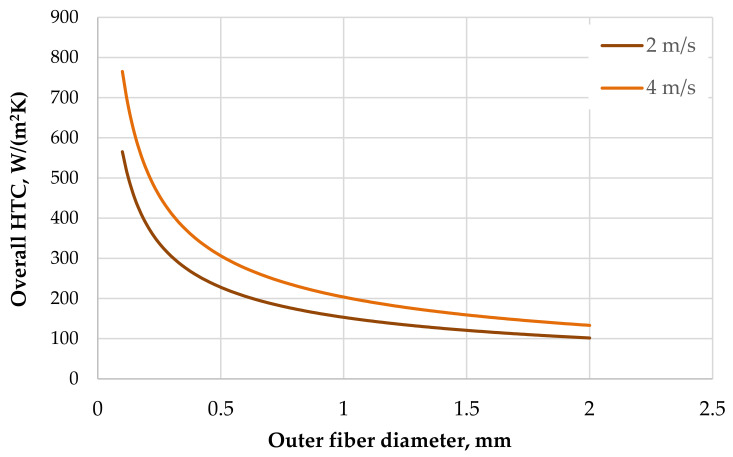
Computed overall heat transfer coefficient for fibers of a variable diameter.

**Figure 3 polymers-13-01175-f003:**
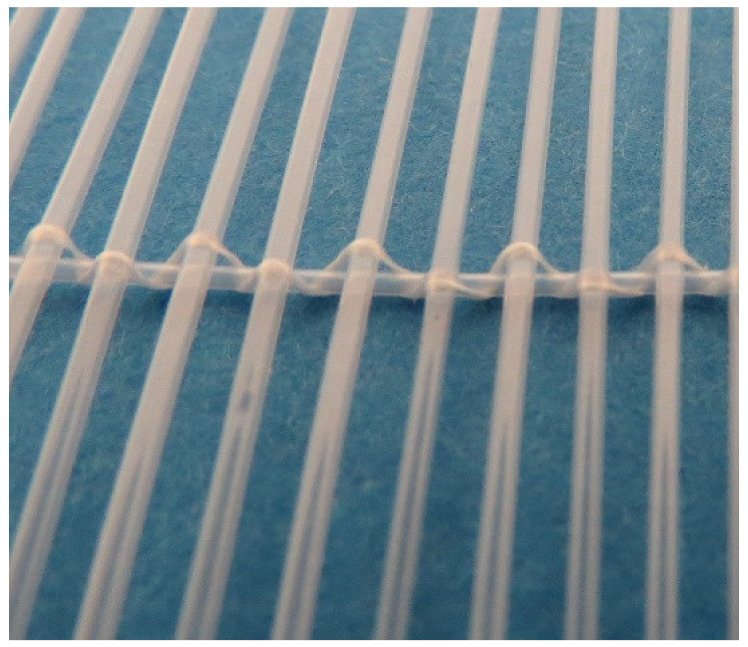
Details of the heat transfer surface used in the novel radiator, separation PP tube, and textile weaving. These tubes are from a different material (Polyamide 11) than the one used in the presented radiator.

**Figure 4 polymers-13-01175-f004:**
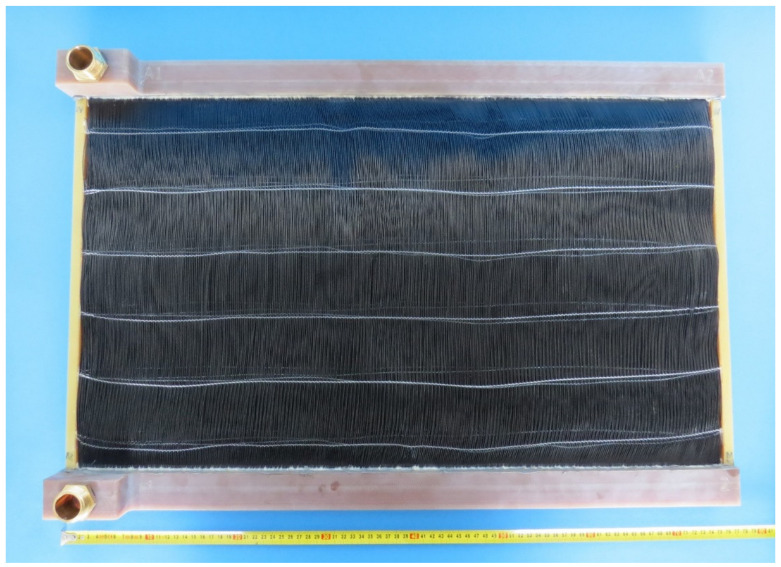
Complete polymeric car radiator used in testing.

**Figure 5 polymers-13-01175-f005:**
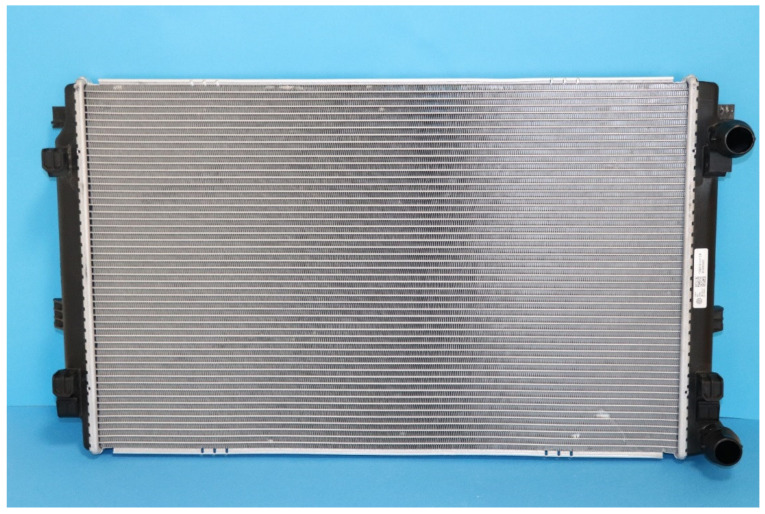
Aluminum radiator from Skoda Octavia used for comparison tests.

**Figure 6 polymers-13-01175-f006:**
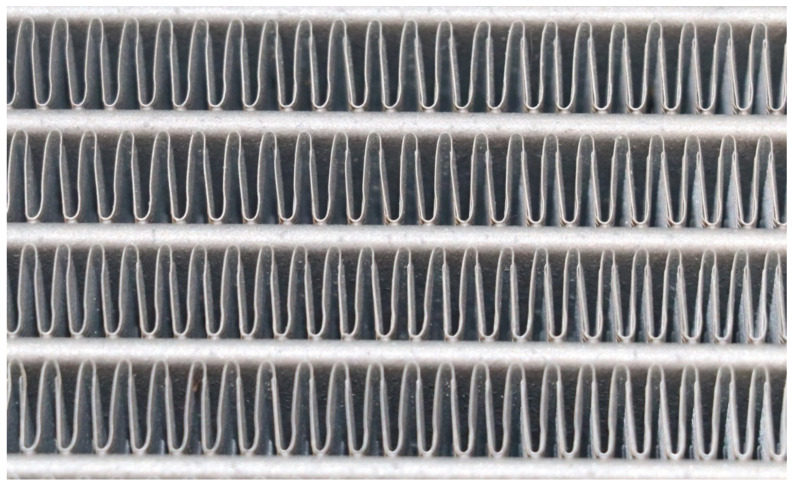
Detail of the heat transfer surface of the aluminum radiator (formed via CAB—controlled atmosphere brazing).

**Figure 7 polymers-13-01175-f007:**
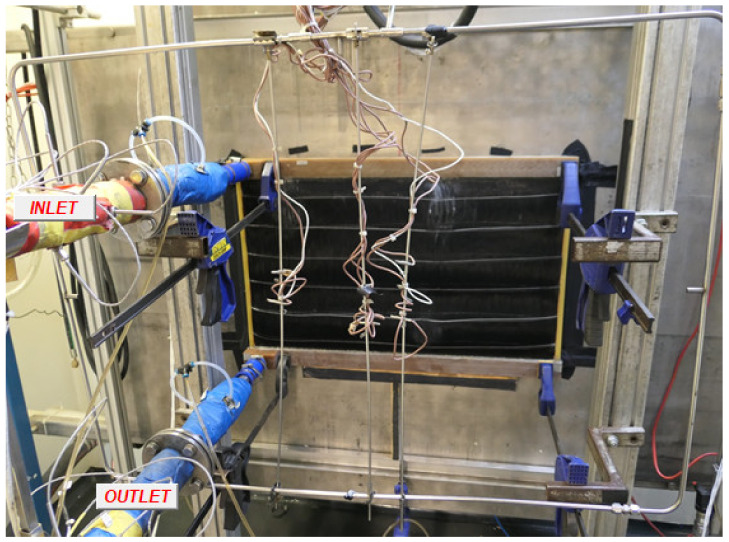
Polymeric radiator in the test circuit where the wires in the front are connected to the temperature sensors.

**Figure 8 polymers-13-01175-f008:**
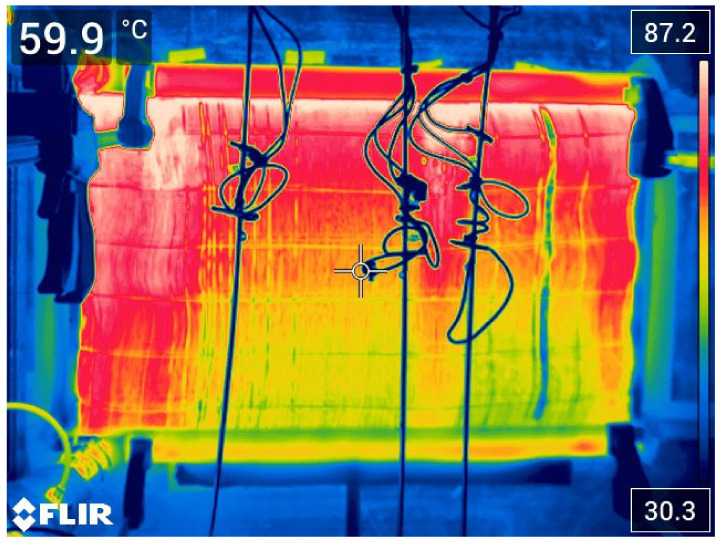
Surface temperature data of the polymeric HX with an intake air temperature of 30 °C, air velocity of 4 m/s, and input liquid temperature of 90 °C.

**Figure 9 polymers-13-01175-f009:**
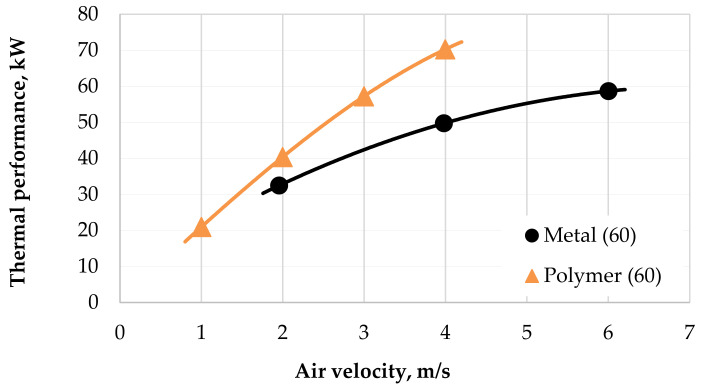
Performance of radiators with variable air velocity and a coolant flow rate 60 L/min.

**Figure 10 polymers-13-01175-f010:**
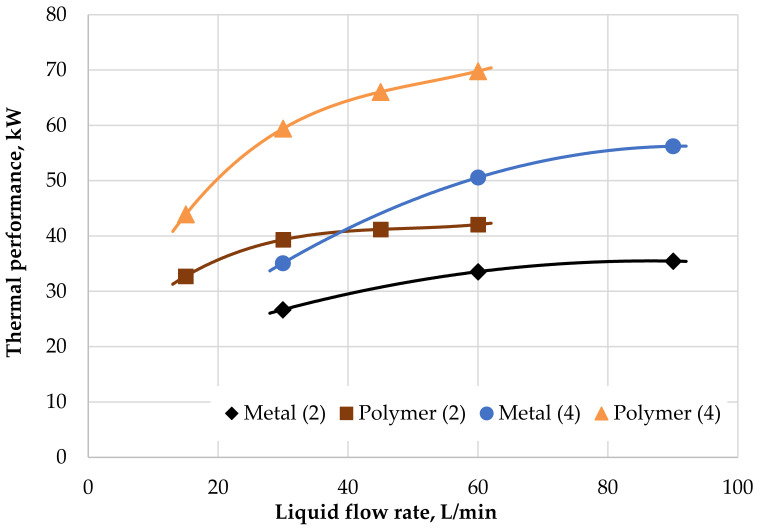
Performance of radiators with variable liquid flow rates and air velocities of 2 and 4 m/s.

**Figure 11 polymers-13-01175-f011:**
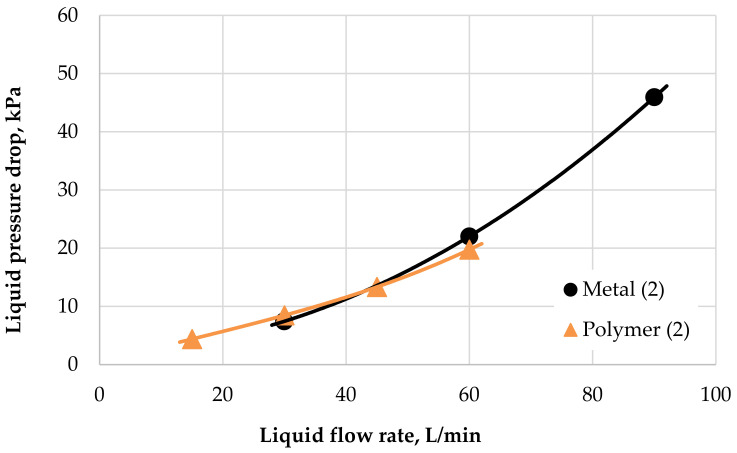
Pressure losses on the liquid side for the radiators, air velocity 2 m/s.

**Figure 12 polymers-13-01175-f012:**
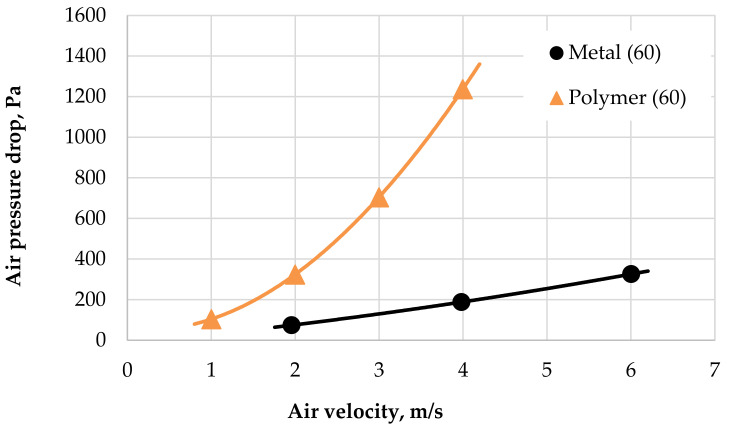
Pressure losses on air side of radiators for coolant flow 60 L/min.

**Figure 13 polymers-13-01175-f013:**
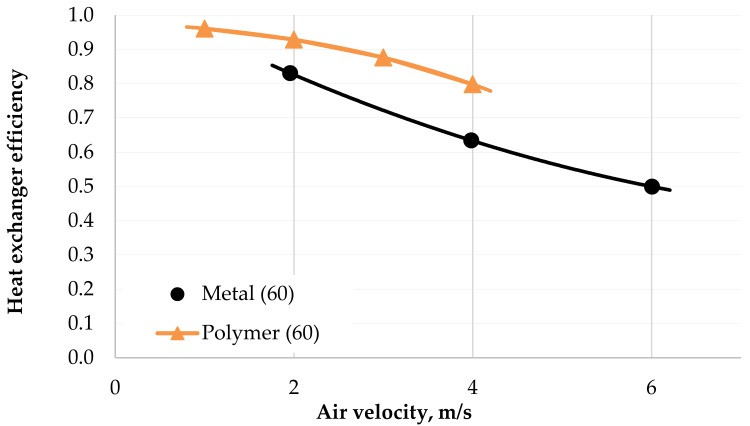
Comparison of the efficiencies of the tested radiators for coolant flow 60 L/min

**Table 1 polymers-13-01175-t001:** Comparison of the two tested radiators.

Type of Radiator	Weight of the Heat Transfer Surface When Dry	Weight of the Heat Transfer Surface with Liquid	Tube Spacing	Inner Tube Size	Wall Thickness
Polymeric	1125 g	2973 g	2 mm	Diameter 0.64 mm	0.08 mm
Metal	2991 g	4218 g	7.8 mm	1.23 × 25.3 mm	0.28 mm

## Data Availability

The data presented in this study are available on request from the corresponding author.

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
