# Peer review of "Comparison of a Novel Polymeric Hollow Fiber Heat Exchanger and a Commercially Available Metal Automotive Radiator"

_polymers, 2021, doi:10.3390/polym13071175_

Round 1
Reviewer 1 Report
This work studies two different types of car radiators, a polymeric and a aluminium one, and author claim that their work compares the functionality and the performance parameters of each radiator type. This work is marginally within the scope of the journal since the polymeric material itself is not studied in the work. Additionally, the exact polymer used is not mentioned in the study, but an unknown polymer is evaluated through its performance in a final product.
Title is confusing and needs to be rephrased.
Language needs improvement throughout the manuscript. The use of a professional proofreading service is advised.
Comparing two commercial products is of no interest for a scientific journal.
At the end of the introduction section, what is studied, how and what it was found is not presented.
Although the references are adequate in the study, the contribution to the field of this work is not clear, and it is not presented and mentioned in the work.
Although the experiment parameters are presented in section 4, a methodology section is missing from the work, making hard to understand the approach of the authors in the evaluation of the two radiators. What is studied, why and how is not adequately presented in the work.
How the heat transfer theory in section 2 was used in this work is not clear afterwards in the manuscrtipt?
How the results of this work were determined is not presented in detail.
The manuscript reads like a technical report with no actual discussion about the results, why this outcome is expected or not, no evaluation with literature and no analysis or even comparison between the results determined in the experiments. If two different radiators, a polymeric and an aluminium one, were compared, how the results would differ? What is the contribution of this work to the corresponding scientific field?
Additionally, the following comments need to be considered, prior to submitting this work elsewhere:
l12 is confusing. Having the overall same dimensions does not mean it is identical. Also, which one the automobile mentioned is using is not clear. If the two radiators were used in different models, two different things are compared.
l44 what "perfect resistance" means? Terms such as this one should be avoided, and the argument need to be described in detail.
l121 how this graph was determined is not mentioned in the manuscript. Same with figure 2.
l155 what was the source of the data in this paragraph.
l160 for the first time here the polymeric material is mentioned to be "polypropylene", but which one, from which vendor and are they any additives in the polymer? This is critical information for the study. How the results would differ, if a different polypropylene, a composite, or a different polymer was used?
l164 what it is shown in figure 3 is confusing. Is this from the actual radiator? Who is the vendor of the radiator? Which area is this?
l172 "heat transfer surface of the tested radiator from Octavia is shown in Figure 7" is this correct?
l186 where did this data were acquired from? A reference should be added.
l195 "test bench is shown in Figure 8." this is not what figure 8 shows.
l218 how the thermal performance in KW was determined in figure 9, is not presented in the manuscript. Same with figure 10, 11, 12, 13.
l253 "Coolant flow inside the fibers is laminar causing the advantage of linear (not parabolic) dependence of pressure loss on the flow rate" this should further explained.
l254 figure 11 is cited in the text after figure 12 (l229), this should be corrected.
Reviewer 2 Report
Figure 2, The caption for the 2 and 4 m/s, referred to the line and to the points is repetitive. Please use only one.
Line 197. The authors refer to figure 11 when surely the correct one is figure 9.
Revise the format of lines 276 to 280.
Then a question. The polymeric radiator seems to show good performance in front of aluminum ones. This is good and in line with the lightweighting strategy of the automotive industry. Nonetheless, to my knowledge the radiators were tested under normal conditions. Do the authors have in mind the test of the radiators under extreme conditions, like a failure of the bypass valve and the creation of vapor?
Anyhow, having in mind the electrification of the automobiles and the need of radiators to control the battery temperature, the presented radiators are a valuable option.
Round 2
Reviewer 1 Report
The revised version of the manuscript is significantly improved in its technical aspects. Several critical errors were corrected, making the manuscript clearer to the reader.
Most of this reviewer’s comments were adequately replied, while others such as, among others, the influence of the specific material on the results of the study, which is a critical comment, were not discussed.
The manuscript in its current form is not submitted with track changes, making the tracking of the changes in the manuscript difficult, since the lines in which these changes were made is not mentioned in the authors reply.
Several arguments of the authors in their reply, such as the importance of their work and others, should be also highlighted in a better way in the revised version of the manuscript.
